# OpenReview forum: "The Road Less Scheduled"
_NeurIPS.cc/2024/Conference — NeurIPS 2024 oral_

### Official Review · Reviewer_8hsg · 2024-07-04

**Soundness:** 3
**Presentation:** 3
**Contribution:** 4
**Rating:** 10
**Confidence:** 4

**Summary:**

This paper proposes a new optimization procedure for deep learning that yields good any-time performance instead of requiring learning rate schedules that decay to zero. The method can be performed on top of standard optimizers, making it easy to apply to existing networks. The paper theoretically describes the method, proving desirable properties. The method is evaluated across a broad range of tasks, including various deep learning tasks (supervised learning) and simpler convex optimization tasks, typically roughly matching or slightly exceeding the performance of standard baselines. The any-time performance of the method in a single run is comparable to the pareto frontier of standard methods when using different schedule lengths, a very large advantage of the proposed method.

**Strengths:**

- The problem studied is of significant relevance to the community and likely to have a large impact. The use of learning rate schedules is almost universal and this has the potential to change that. Strong anytime performance can significantly simplify experimental procedures.
- Good theoretical contribution.
- Strong empirical evaluation across a large number of tasks with sufficiently strong baselines with very good results.
- Significant novelty as far as I know, I have not seen any works that can provide this type of any-time performance before.
- Paper is relatively easy to follow with clear figures.
- Experimental validation is done over multiple seeds.

**Weaknesses:**

- Deep learning evaluation could be improved slightly, especially with respect to hyperparameter tuning and transferability of the hyperparameter values. The effects of different hyperparameters and how they change between settings is not discussed. The full ranges of the hyperparameter tuning of the method and baseline is not provided.

**Questions:**

Minor Suggestions:
- L20: Specify whether g_t is evaluated at x or z
- L29: Maybe clarify that this “T in advance” applies to many popular schedules. There are many examples of schedules that don’t have this, e.g. trapezoidal schedules / warmup-stable-decay / reduce-on-plateau.
- L41: No additional hyperparameters over SGD *with momentum* or Adam
- Maybe include equation numbers, they make it much easier for people to refer to the equations e.g. when discussing the paper even if you do not use these references in the manuscript. Labeling equation 1 as such seems odd since nothing else is labeled.
- L178: Why is Lookahead like EMA? I can see the resemblance to PA but not necessarily EMA.
- L290: It would be nice to show some results for different types of weighing when using a warmup

**Limitations:**

Yes, no concerns here.

---

> ### Author Rebuttal · Authors · 2024-08-05
>
> Thank you for providing line-by-line comments we will update the camera ready and arxiv to reflect all of your suggestions. We also address them below
>
> L20: Specify whether g_t is evaluated at x or z
>  - Good point, we will update.
>
> L29: Maybe clarify that this “T in advance” applies to many popular schedules. There are many examples of schedules that don’t have this, e.g. trapezoidal schedules / warmup-stable-decay / reduce-on-plateau.
>
>  - Yes, we will state that there does exist schedules that instead require a cool-down period to be chosen, which can be chosen during the course of the run, and that the historically popular reduce-on-plateau schedule also doesn't require a stopping time (with the downside that it is significantly out-performed by modern cosine/linear schedules).
>
> L41: No additional hyperparameters over SGD with momentum or Adam
>  - This should specify SGD+momentum not just SGD, we will fix.
>
> Maybe include equation numbers, they make it much easier for people to refer to the equations e.g. when discussing the paper even if you do not use these references in the manuscript. Labeling equation 1 as such seems odd since nothing else is labeled.
>  - We will add equation numbers through the whole paper. Thanks!
>
> L178: Why is Lookahead like EMA? I can see the resemblance to PA but not necessarily EMA.
>  - By EMA here we mean that the x sequence is an EMA, since it's updated with a fixed interpolation weight $\alpha$ rather than a decreasing weight sequence that would given an equal-weighted average. This can be seen by rearranging the equation for $x$ in the more classical EMA form: $x_{t}=\left(1-\alpha\right)x_{t-1}+\alpha z_{t,k}$. It's not strictly an EMA of the z sequence just of the last z from each inner loop. We will clarify this in the paper.
>
> L290: It would be nice to show some results for different types of weighing when using a warmup.
> This is a good suggestion. During the early development of the algorithm we did a sweep of weighting value powers and found that a power 2 works the best. We have not performed a large scale ablation of this value though. We will add this to the camera ready.
>
>
> We are very glad to see your enthusiasm for our method! We want to note that a submission to the AlgoPerf competition that used our method has achieved a first-place result in the final standings, providing further independent evidence of the practicality of our method. We hope you will consider increasing your score in light of this.

---

> > ### Comment · Reviewer_8hsg · 2024-08-08
> >
> > I thank the authors for their response and the additional hyperparameter sensitivity sweeps. I think the additional experiments and proposed changes improve the paper further, and the AlgoPerf results are very impressive as well. Overall I think this is a probably a top 10 paper at this conference, I will raise my score to reflect this.

---

### Official Review · Reviewer_mxnW · 2024-07-09

**Soundness:** 3
**Presentation:** 4
**Contribution:** 4
**Rating:** 8
**Confidence:** 3

**Summary:**

The paper proposes a scheduler free method for training, analyzes it theoretically to show that it matches the theoretical benefits of Polyak averaging while recovering the performance of standard cosine decay used in practice. Their approach can be interpreted as being an interpolation of Polyak averaging and Primal averaging (which is equivalent to momentum). Thus it gets the benefits of acceleration while maintaining the low variance of Polyak averaging.

**Strengths:**

1. The empirical comparison to cosine decay scheduler is done in a thorough manner.
2. Another strength of this work is that alternative approaches such as Polyak averaging or EMA require maintaining one extra copy of the weights.

**Weaknesses:**

Since the focus of the paper is to match theoretical guarantees of Polyak averaging while matching empirical performance of schedulers they also need to compare to Polyak averaging and EMA variants[1]. The paper does not do this thoroughly.
1. Line 22 states “Despite their theoretical optimality, PR averages give much worse results in practice than using the last-iterate of SGD with well-chosen learning rate sequences (Figure 2a)” but Figure 2a states the Polyak-averaging diverges. Clearly this could be fixed with a slightly smaller learning rate since cosine decay did not destabilize?
2. The paper does not compare to EMA which is another method which can empirically reduces variance while not using schedules.

**Questions:**

Could the authors add comparison to (tuned) EMA? I would happy to increase my rating further if the authors do this.

**Limitations:**

Yes.

---

> ### Author Rebuttal · Authors · 2024-08-05
>
> # Weaknesses
>
> 1.
>
> This is a really good point. It is possible to get Polyak averaging to converge by using a smaller LR value. For the IWSLT14 illustrative plot we did not include a full LR sweep (as we did with all experiments in the experiments section) when we should have. We have ran this LR sweep and included the updated plot in the global rebuttal PDF. It shows that both Polyak and Primal averaging still greatly under-perform compared to both a cosine schedule and the Schedule-Free approach.
>
> We currently do include Polyak and Primal averaging in all convex-case experiments but we can also run these for the deep learning experiments for completeness. We will run these additional experiments for the camera ready. After reflecting on your question, we realize that there hasn't been any published experiments extensively comparing Polyak or Primal averaging to schedules in the non-convex setting, and their sub-optimality is more a folk-law result. These additional experiments could be useful to the community.
>
> 2) See experimental results below.
>
> # Questions
> *Could the authors add comparison to (tuned) EMA?*
> This turned out to be an interesting question. We ran an experiment on CIFAR-10 using SGD with momentum 0.9, using the same experimental setup as in the paper, and with warmup-then-flat learning rate schedule.
> We found that with a careful sweep of the exponential model averaging parameter, we are able to achieve essentially the same test accuracy as we get with Schedule-Free. See below a list of EMA values and the associated test accuracies (single seed), sorted from best to worst:
>
> 0.9998: 96.02
>
> 0.9996 95.92
>
> 0.9999: 95.83
>
> 0.99996: 95.743
>
> 0.998: 95.66
>
> 0.999: 95.60
>
> 0.99998: 95.343
>
> 0.996: 95.313
>
> 0.99: 94.852
>
> 0.99999: 92.768
>
> This is an interesting result! It suggests that the averaging in Schedule-Free may have a similar effect to exponential weight averaging, but without the requirement to tune an additional parameter, and also without the additional memory cost of EWA. Thank you for the suggestion that we investigate this. This link definitely warrants further investigation and we will run additional experiments for the camera ready that directly compare to EMA.
>
> We also tried running exponential model averaging using a warmup+cosine schedule for the SGD+M method. This was significantly worse across the board than the warmup+fixed schedule, which is surprising as existing papers usually stack EMA with a schedule. See the results below:
>
> 0.998 95.48
>
> 0.99996 95.45
>
> 0.9996 95.38
>
> 0.9998 95.38
>
> 0.95 95.33
>
> 0.98 95.30
>
> 0.999 95.292
>
> 0.996 95.24
>
> 0.99 95.16
>
> 0.9999 95.04
>
> When combining with a schedule there is less sensitivity to the choice of $\beta$.
>
> We hope these experiments help answer your questions. Please let us know if you have any further questions or comments.

---

> > ### Comment · Reviewer_mxnW · 2024-08-07
> > **Response to authors**
> >
> > Thanks for these experiments. Since the experiments with EMA indicate that EMA can potentially match the performance of ScheduleFree I will maintain my score instead of increasing it.

---

> > > ### Author Response · Authors · 2024-08-07
> > > **Response**
> > >
> > > Thanks for the quick response during the discussion period, we appreciate it. This EMA approach is interesting, and we want to note a few things:
> > >
> > >  - The idea of using EMA as a replacement for a schedule has not appeared in the literature before as far as we are aware, and when used with a decreasing schedule as is normally done, we see that the EMA doesn't match Schedule-Free performance.
> > >  - The EMA requires precisely tuned momentum of 0.9998, it underperforms with similar momentum values such as 0.9996 and 0.9999. The momentum value also depends on the length of training, so does not maintain the nice properties of Schedule-Free learning. This extreme hyper-parameter sensitivity would make it very difficult to use in practice.
> > >  - The EMA sequence requires an additional memory buffer over Schedule-Free as the base optimizer must also use it's own momentum to prevent divergence. This is not necessary with Schedule-Free learning as the interpolation operation stabilizes the iterate sequence.
> > >
> > > So overall, EMA doesn't have the same low-memory and low-tuning properties of Schedule-Free learning. Based on those notes, we hope you will reconsider raising your score.

---

> > > > ### Comment · Reviewer_mxnW · 2024-08-08
> > > > **Response**
> > > >
> > > > Thanks for your reply. I have increased the score.

---

### Official Review · Reviewer_Q984 · 2024-07-12

**Soundness:** 2
**Presentation:** 3
**Contribution:** 3
**Rating:** 5
**Confidence:** 4

**Summary:**

The authors propose a method for training neural nets without needing to know the total training time T in advance. This contrasts with a standard training setup where one chooses a learning rate schedule in advance, and the schedule must include an a priori chosen stopping time T (e.g. cosine schedule or linear decay). If the method really works as advertised it therefore simplifies practical training setups and reduces the cost of training, by avoiding the need to run multiple training runs with different values of stopping time T.

The method works by tracking two sequences: one is the "noisy" sequence z that integrates noisy gradient updates, the second is the "smoothed" sequence x that tracks a uniform average over all past noisy iterates z. Gradients are evaluated at an interpolation between the current noisy z and smoothed x sequence, and added back to the noisy z sequence which is then averaged into the smoothed x sequence.

Authors present theoretical results about their technique for convex optimisation, and evaluate the method across a set of training benchmarks, some that involve tuning and some that don't. The results look promising.

**Strengths:**

- the idea is legitimately really cool and clever and seems very original to me. People know that weight averaging on top of standard training can boost performance, so it's really cool to ask "is there a way to fold this back into the training loop" as the authors do.
- my broad understanding of the technique is that it's doing noise filtering in a clever way, and I think that this idea can inspire others to explore this direction, which is great
- the paper is very clearly written and I like the mix of the intuitive description of the technique, the formal results, and then the experimental evaluation
- it's great that the authors engaged with the MLCommons Algorithmic Efficiency benchmark, and from what I understand the results are very promising.

**Weaknesses:**

Okay as I've mentioned, I think the idea is cool and original and can inspire followup work, which is all we really want of a paper. So I'm going to focus on giving you feedback which is intended to be constructive and I hope can help you generally improve the quality of the work or followup work that you do. I'm not going to hold the paper hostage, but if I feel like you engage meaningfully with my critiques I'm willing to upgrade my score. All of my critiques can be addressed either by making minor amendments to text or by adding a limitations section at the end of the paper *or* by running more experiments and addressing them, but I won't insist on which and leave this up to the authors.

### **Paper may be over claiming slightly on its results**

The paper makes claims to be **"one of the largest and most comprehensive machine learning optimization algorithm evaluations”**. Can you quantify in what sense this is true? To me the evaluation doesn't feel that comprehensive---for instance, there is only one plot on hyperparameter sensitivity, while a broad swathe of the community has started to explore this question as standard in evaluations papers. The authors also state their method **"has no notable memory, computation or performance limitations compared to scheduling approaches"**. In my opinion, the truth value of this statement is essentially unknowable without more thorough evaluation. It would be fine to amend the statement by saying "WE BELIEVE THAT our method has no notable memory, computation or performance limitations compared to scheduling approaches."

### **Hyperparameter sensitivity is not thoroughly investigated**

In my opinion, the main potential limitation of the work is that tuning hyperparameters (learning rate, weight decay, interpolation constant $\beta$) may implicitly be tuning an LR schedule. This is especially the case for this technique since the role of these hyperparameters is quite subtle given the unusual form of the update sequences. Now the authors may argue that they address this with their MLCommons Algorithmic Efficiency experiments that reportedly use the same set of hyperparameter across all tasks, but in my opinion this could be a fluke relating to all the MLCommons tasks involving similar training times for instance. What I would want to see to convince me otherwise is to see experiments that sweep across beta for different training times, and check if the optimal beta is invariant to training time, say.

### **Paper applies a potentially ill-suited theoretical framework to deep learning, and as such there are gaps**

The paper comes from a part of the community that uses convex optimization frameworks to do algorithm design, and then extends the methods to deep network training. Sometimes the extension can feel a bit forced: e.g. let's switch to Adam instead of SGD as our base optimiser because it works better, let's use learning rate warmup because it works better, etc. The paper uses theorems within this convex framework to support its significance, but as far as I can tell these theorems don't answer basic questions a practitioner would have about the technique: e.g. Theorems 1 and 2 provided no guidance on setting the interpolation parameter $\beta$ since they hold uniformly for all $\beta \in [0,1]$.

I want to point out that another part of the community is trying to build up understanding and frameworks for deep learning fundamentals that involve less of a "jump" from convex optimisation to deep learning. To point out two examples:
- https://arxiv.org/abs/2103.00065 "Gradient Descent on Neural Networks Typically Occurs at the Edge of Stability"
- https://arxiv.org/abs/2405.14813 "Scalable Optimization in the Modular Norm" (concurrent work)

I think it would be great to at least read these works and potentially engage with them---not in this paper, but in the future

### **Final note**

I sometimes worry about giving frank feedback as I think I can sometimes have a blunt style. I want to re-emphasise that I really like this paper. I think the idea is clever and creative, and I really encourage you to pursue these directions further. The feedback is intended constructively. I am currently giving the paper a score of 5 because a score of 6 requires "no major concerns with respect to evaluation" and I am not there yet. I can get there if I become confident that you've engaged with my review.

**Questions:**

- do you know if beta is sensitive to e.g. training time? Fine if you don't know, but consider adding a limitation saying future work could investigate this closer
- which sequence is used to do inference x, or y, or z? It might be worth clearly flagging this in this paper. Sorry if I just missed it.

**Limitations:**

"Note that we excluded one benchmark problem, ResNet-50 training, as neither AdamW nor NAdamW can hit the target accuracy on that task." ---- this feels artificial to me. I still want to know how schedule free AdamW does!

---

> ### Author Rebuttal · Authors · 2024-08-05
>
> # Strengths
> We are glad that you find our work interesting! We think that this approach has wide applicability and we are doing our best to spur adoption by doing an open source release in both PyTorch and Jax. An entry using our method was entered into the AlgoPerf competition earlier this year, and the final results were just released this week, showing that Schedule-Free AdamW ranked **first** in the self-tuning division, beating out all other entries including the AdamW+Cosine implementation tuned by the organizers. We hope this independent verification will help you assess the potential impact of our work.
>
> # Weaknesses
>
> ## Paper may be over claiming slightly on its results
>
> *"one of the largest and most comprehensive machine learning optimization algorithm evaluations”. Can you quantify in what sense this is true?*
>
> We will reword this phrase as we agree it is not precise enough. We wanted to convey that we run experiments on a larger set of test problems than any prior paper on deep learning optimization. As far as we are aware, there is no paper out there that runs as many full-training (not fine-tuning) experiments as we do on deep learning problems. The are larger comparisons on logistic regression and other convex problems in past literature, as well as model fine-tunings, so we need to be careful to adjust our wording to clarify.
>
> We have ran additional hyper-parameter sensitivity experiments, please see the section below.
>
> *"has no notable memory, computation or performance limitations compared to scheduling approaches"*
>
> - We will cut this phrase from the conclusion as we agree it is overreaching at the moment.
>
> ## Hyper-parameter sensitivity is not thoroughly investigated
> This is a good point, we don't currently investigate hyper-parameter sensitivity thoroughly. We ran a series of additional experiments and have included these results in the PDF attached to the global rebuttal. We ran as many experiments as we were able to in the 1 week rebuttal period, so they are not completely conclusive. We will run additional experiments for the camera ready.
>
> ### Learning Rate
> We ran a series of learning rate sweeps for the smaller problems in our test bench (these sweeps are time consuming to run), CIFAR10, CIFAR100 and SVHN. We see that the sensitivity to the optimal learning rate is very similar to SGD with momentum, with some variability observed. For CIFAR10/100 the curves look essentially the same, and for SVHN the curve has a somewhat tighter peak.
>
> ## Momentum
> For our momentum sweep, we see a similar result. For both problems that we ran, neither Schedule-Free or the SGD+Momentum baseline show clearly more or less sensitivity to the hyper-parameter, and there isn't a clear pattern.
>
> ### Beta v.s. Training Time
> See the Questions section below.
>
> ## Paper applies a potentially ill-suited theoretical framework to deep learning, and as such there are gaps
> The analysis framework to use here is a question we constantly struggle with as researchers in this area. The convex framework is usually considered ill-suited to describe deep learning problems, but it leads to results that seem to work well in practice (this paper's method is a great example). Averaging of iterates for general non-convex problems can't be analyzed as far as we are aware, since you can end up averaging between winding valleys of the parameter space and the resulting points don't necessarily have low loss.
>
> In terms of the optimal values of $\beta$, we are investigating this as followup work. So far we have a result that for stochastic quadratic problems $\beta=0.5$ is optimal. A similar result holds for strongly convex problems although it may not be strictly optimal in that case. In other convex settings, larger $\beta$ values than 0.5 are potentially better, but there is a dependence on the local quadratic-ness of the problem, and the more quadratic, the closer to 0.5 the value should be.
>
> ## Questions:
>
> 1) *do you know if beta is sensitive to e.g. training time?*
>
>  We have run an additional experiment to understand this dependence for the ImageNet test problem, it is included in the global rebuttal PDF. We ran training out to 200 epochs instead of 100, to see if larger $\beta$ values given improvements at the very late-stages of training.
>
> We find that at the early stages of training, the value of $0.9$ gives the highest test accuracy, but at approximately epoch 50, the larger value of $0.95$ starts to dominate (we didn't run $0.9$ in our sweep in our paper, so this result is an improvement over the previous results). This $0.95$ value dominates for the remainder of training, and the larger value of $0.98$ is far behind in terms of test accuracy. The beta=0.75 run doesn't do better at the beginning, so it's not the case generally that smaller beta is always better at the beginning.
>
> So to summarize, smaller $\beta$ values can perform better for shorter duration training runs, but this dependence on training time for the optimal beta seems very mild.
>
> 2) *which sequence is used to do inference x, or y, or z?*
>
> This a good suggestion, we don't currently clearly indicate which sequence is used. We will update the paper. The sequence used for inference should be the $x$ iterate.
>
>
> If you have any further questions please don't hesitate to ask.

---

> > ### Comment · Reviewer_Q984 · 2024-08-09
> >
> > Thank you for your rebuttal. I'm just responding with my thoughts to provide a chance for more discussion:
> >
> > **"Schedule-Free AdamW ranked first in the self-tuning division"** Congrats on this. I'm not 100% sure how competitive this track was (5 submissions, 4 of them struggling). But still it's really impressive you beat the baseline set by the organizers, so fair play! And again I already mentioned in my review I wasn't sure what the diversity of benchmarks was in terms of problem scale.
> >
> > **"We ran as many experiments as we were able to in the 1 week rebuttal period"** I appreciate that it's going to be difficult to get a lot done in one week, so thank you for running these extra experiments. Still these experiments are not quite what I had in mind. **I think the paper would strongly benefit from running these sensitivity sweeps at a variety of problem scales, e.g. transformer training on 10k, 100k, 1M, 1B tokens and check whether the minimum of each sweep lines up or not.** Sweeping training time on a log grid here I think is pretty crucial. Also varying number of tokens instead of number of ImageNet epochs avoids confounding factors from doing multiple passes over the same data. But thanks for running the ImageNet one---and I suppose it is showing some dependence of optimal beta on training time.
> >
> > **"The convex framework is usually considered ill-suited to describe deep learning problems, but it leads to results that seem to work well in practice"** Again I'll be a little blunt, but please understand I'm just trying to convey my opinion and then we can discuss. From my perspective, it seems like the convex opt stuff is a great source of inspiration in your work, driving you to test novel algorithms that no one else is thinking of. This is amazing. However, I'm not recalling any evidence in your paper that shows that the theory you develop has any connection to how your method actually works in practice. I'm a bit suspicious about this aspect.

---

> > > ### Author Response · Authors · 2024-08-09
> > > **Response**
> > >
> > > This is a good point. There is a significant difference in scale between the problems in AlgoPerf, ranging from solving in minutes to days, but it's not at the level of modern large-scale training runs, as the entire benchmark suite is designed to run beginning-to-end in under a week on 8 V100 GPUS (2 generations old!).
> > >
> > > We are also very interested in evaluations at larger scale as this is crucial for wide adoption of our method. We plan to run these evaluations over the coming months. These evaluations are time-consuming as the optimal LR values, momentum and decay for Schedule-Free differ from schedule-based runs, and so we need to run large log-spaced parameter sweeps.
> > >
> > > *"The convex framework is usually considered ill-suited to describe deep learning problems, but it leads to results that seem to work well in practice"*
> > >
> > > Most researchers in this area take a very different approach than we do. We want to elaborate on this further as it's central to the development of this method.
> > >
> > > It's clear that non-convex deep learning problems don't inherit many of the nice-properties of convex problems. For example, it's often the case that methods that rely on estimating smoothness fail when extended naively to the deep learning setting. However, there is a growing belief in the community that the *online* convex optimization framework, which only assumes bounded gradient rather than smoothness, can accurately model the behavior of non-convex learning. This was captured in a recent COLT Open Problem submission: https://www.ehazan.com/open_problem.pdf. Any progress on this open problem, developing a theoretical black-box reduction between the two settings, would justify our method.
> > >
> > > So as you say, we don't directly answer the question of why our method works in the setting of our deep learning experiments. We don't know how to answer this question concretely yet. It's wild that it works - averaging of iterates makes little sense on non-convex problems, and so some level of local convexity must be present, and seemingly far more than we would have believed before we started this line of research! We are actively researching this ourselves, and we hope that our work also spurs others to look into this further.

---

> > > > ### Comment · Reviewer_Q984 · 2024-08-10
> > > >
> > > > Hi---thanks for the reply!
> > > >
> > > > **"These evaluations are time-consuming"** Agreed and I hope I made it clear in my initial review I wasn't asking you to do this during the rebuttal. What I want is for you to flag these limitations clearly in the paper so that it's easier for readers to have a well-calibrated sense of your results. If you don't make these limitations clear, I would consider the paper to be over-claiming. The crucial point that I want to make absolutely sure that you're getting is that **if you have to tune HPs for different length training runs, then the method is effectively not schedule-free**. Feel free to argue against this, but if not I need to see an acknowledgement that you understand my point, and a commitment to flagging this as a limitation.
> > > >
> > > > **"It's clear that non-convex deep learning problems don't inherit many of the nice-properties of convex problems."** The perspective I have here is that deep learning problems have their own nice structural properties that we need to characterize. And again, researchers are making efforts to do this.

---

> > > > > ### Author Response · Authors · 2024-08-10
> > > > > **Follow up response**
> > > > >
> > > > > We fully understand the motivation for your request here. Since our results include sweeps of LR, decay and momentum, without evidence that these parameters are not heavily dependent on the training run length, than we could just be "shifting" the need to set a duration-dependent schedule into the need to set a duration dependent LR/decay/momentum. We will run more comprehensive experiments for the camera ready demonstrating that the optimal hyper-parameters are not heavily dependent on the run duration. We will devote a section to this, as this is an important question that our work currently does not answer. This will involve sweeping each parameter at different run durations, to determine if shorter or longer runs require different hyper-parameters for optimal performance, both for schedules and for Schedule-Free learning.
> > > > >
> > > > > These results will be interesting as we are not aware of any existing comprehensive examination of hyper-parameter dependence on training duration currently in the literature even for schedules. Thank you for this suggestion!
> > > > >
> > > > > Regarding the analysis framework, we see our approach as complementary rather than in competition with such work. In particular, theory-work that more directly addresses the properties that neural networks possess that make training tractable for them is extremely important and we follow the literature closely there. It's clear that there is a lot going on that is not captured by the classical smooth-non-convex analysis framework that is often used in the optimization literature. We hope that bridging this divide via reductions that apply for certain restricted problem classes (as described in the Open Problem mentioned above) will help bring us closer to developing optimization methods that better exploit the properties of neural networks. It's an exciting area!

---

> > > > > > ### Comment · Reviewer_Q984 · 2024-08-10
> > > > > >
> > > > > > Thanks for this, and I appreciate you clarifying that we're on the same page. I hope you don't take my top level comment above as an affront and I think / hope the paper will be stronger for the criticism. I agree that you've worked out a really cool method, and I actually think it may be useful even for orthogonal reasons to whether or not it's truly schedule free.
> > > > > >
> > > > > > I also think you will / you have already inspired a lot of cool followup work. I just want to try to help bring the paper into a form where the community can make as efficient progress as possible

---

> > > > > > > ### Author Response · Authors · 2024-08-10
> > > > > > > **Fantastic discussion**
> > > > > > >
> > > > > > > Thank you, we really do appreciate your candid discussion. We think that our work is greatly improved by incorporating your suggestions and we hope that you see it that way also, even considering your concerns about our work. We hope that you will consider raising your score in light of our back-and-forth discussions to reflect your overall positivity about our work.
> > > > > > >
> > > > > > > Thank you again!

---

### Official Review · Reviewer_rKis · 2024-07-13

**Soundness:** 3
**Presentation:** 3
**Contribution:** 3
**Rating:** 7
**Confidence:** 3

**Summary:**

This paper proposed an optimization style for stochastic optimization, termed schedule-free optimization, which is free of manually selected/tuned learning rate schedulers. The proposed method enjoys both the worst-case optimal last-iterate convergence and promising empirical performance. The authors also introduced a general online-to-batch analysis framework to analyze the proposed method. The empirical results show that the proposed method outperforms the state-of-the-art methods in various tasks, including training large language models (LLMs) and image classification tasks.

**Strengths:**

1. The mismatch between the theoretically optimal learning rate schedule and the practice is an important problem. This paper re-introduces and re-emphasizes this issue in a well-educated manner.
2. The proposed online-to-batch analysis framework is general and insightful.
3. The positive result in the empirical verification of equation (9) introduces an interesting new open problem that is worth further investigation: why could well-known convex/non-convex optimization problems as such an "bounded-regret" property under gradient descent?
4. The proposed schedule-free method enjoys both the worst-case optimal last-iterate convergence and promising empirical performance.

**Weaknesses:**

1. In the update rule of $z_{t+1}$, using $y_t$ instead of $z_t$ could incur additional forward passes during the optimization process (of neural networks through backprop), which especially undermines the efficiency and applicability of the proposed method in the scenario of model parallelism.
2. The adaptation from schedule-free SGD to schedule-free AdamW is still in a heuristic way.

**Questions:**

1. Why don't you report any empirical results that combining schedule-free methods and parameter-free methods? By the way, certain parameter-free literature is cited in the Appendix for the use of proof. The authors are encouraged to elaborate on the connection between this citation and the current work.
2. Any conjecture/explanation on the empirical observation on Line 158?
3. How do you deal with the "decoupled weight decay" issue in adapting the most popular optimizer for LLMs (i.e., AdamW) to the schedule-free style?

**Limitations:**

No major societal limitations.

---

> ### Author Response · Authors · 2024-08-05
> **Reviewer 1 Rebuttal**
>
> Thank you for the detailed review, we appreciate it. We are very glad you see the potential of our method. We would like to start by addressing each of your concerns separately:
>
> Weaknesses:
>  1) *additional forward passes* This is a good point! In our PyTorch and Jax implementations, we are able to avoid any additional forward or backward passes over regular AdamW, except when BatchNorm is used, where the extra forward passes are needed to warmup the BN running_mean/var buffers before a test loss evaluation. These extra forward passes have less than 2% runtime overhead. Given this low overhead, and the fact that batch-norm is quickly being replaced by layer-norm and other alternatives in modern architectures, we don't think this is a major issue.
>
> 2) *heuristic adaptation for AdamW* As you say, our Schedule-Free AdamW version is heuristically adapted from standard AdamW. We will add a section to discuss this to the paper. AdamW doesn't have a concrete regret bound, but related methods that "fix" AdamW's theory, such as AMSgrad, can also be used. Since our main Theorem shows how any method with a proven regret bound can be used adapted to be Schedule-Free, it is then no longer heuristic. Our theorem handles arbitrary learning rates including warmup, and any weighting sequence, so this bound directly applies to the methods as practically implemented.
>
> Questions:
> 1) *LR Adaptation* We have done some initial investigation into integrating Schedule-Free with various recently described LR adaptation methods. This integration is surprising subtle as the theory doesn't apply directly for the technical reason that D-Adaptation and DoG methods show bounds on the stochastic loss, NOT the regret, whereas our method specifically requires a regret bound. This difference is crucial to those methods behavior from the theory point of view. We so far have not seen good results when using Schedule-Free in combination with D-Adaptation or Prodigy. We are planning to next investigate integrating with regret-bound methods such as Mechanic and Coin-Betting approaches in the future, and we hope to see good results there.
> 2) *Line 158 - Momentum allows larger LR values* In the quadratic setting, for large beta values, our method becomes convergent for larger learning rates then you would normally be able to use. This is likely related to the similarity between our method and momentum which provides acceleration in the quadratic case. However, we don't believe this result is indicative of the actual behavior of the method on deep learning problems as it only holds in the quadratic setting. We have further theoretical results in this direction that didn't make it into the paper by the submission deadline, and we see in the stochastic strongly convex case, our method provides provably better convergence rate bounds with beta \in (0,1) than is achieved with Polyak or Primal averaging.
>
> 3) *decoupled weight decay* Integration of optimization methods with decoupled weight decay is a surprisingly subtle issue. Currently we have a switch in the open source code that supports weight decay calculated either at $y$ (the default used in our experiments) and $x$. We find empirically that calculating it at $y$ performs a tiny bit better on some problems but it largely doesn't matter. From a theory point of view, you can make arguments for both forms, which is why we support both. We will add additional clarification around this difference in the paper.
>
> We hope that you will consider raising your score given our comments above. We believe this work has potential to become the default training method in the future given it's strong advantages over classical schedules and it's ease of use. An entry using our method was recently announced as the first-place winner in the self-tuning track of the AlgoPerf competition, a further indication of the potential of our method. It significantly beat all other entries including a tuned AdamW.

---

> > ### Comment · Reviewer_rKis · 2024-08-05
> >
> > The authors largely covered my questions, with an interesting one left. I reiterate that point here, which I mentioned in the original review.
> > > By the way, certain parameter-free literature is cited in the Appendix for the use of proof. The authors are encouraged to elaborate on the connection between this citation and the current work.
> >
> > I mean line 620 ("Recall from D-Adaptation ...") in the current version of the paper.

---

> > > ### Author Response · Authors · 2024-08-05
> > > **Response**
> > >
> > > Thank you for the quick response!
> > >
> > > To answer your question, we have adopted the analysis framework developed recently by Defazio and Mishchenko as it makes the proofs of many results that link stochastic optimization and online optimization shorter and simpler. This framework is actually generic and can be applied to most methods for online optimization, not just parameter-free methods such as D-Adaptation. Our theory doesn't directly use any results that relate to parameter-freeness, just the basic online-learning analysis inequality they develop. It makes the dependence on the norm of $s$ more explicit in the bounds compared to classical inequalities such as used in Orabona's Online Learning monograph.

---

> > > > ### Comment · Reviewer_rKis · 2024-08-12
> > > >
> > > > > Thank you for your answer.
> > > >
> > > > - The authors have addressed my quesitons in the "Quesitons" section.
> > > > - The authors have tried to address my concerns in the "Weaknesses" section in a decent way.
> > > >
> > > > Overall, I will keep my positive evaluation unchanged.

---

### Author Rebuttal · Authors · 2024-08-05

Several reviewers requested additional plots to examine the hyper-parameter sensitivity of our method. We have ran as many experiments as time allowed in the rebuttal period, covering the sensitivity to learning rate, momentum and the duration of training.

---

### Comment · Reviewer_Q984 · 2024-08-10
**The paper is interesting and worth presenting, but the reviewer scores are too high**

Dear AC, fellow reviewers, and authors,

I like this paper, and I think it is in the interest of the community to present it at the conference, but I want to highlight what I believe are the major limitations of this paper, since I believe they are currently being overlooked. **I believe that the current reviewer scores are far too high, and I feel strongly about this**. I am deliberately trying to challenge the authors and the other reviewers and I hope that we can have a fruitful discussion about this. **Again, I believe the paper should be accepted but I do not believe that it is award quality.**

**The connections between learning rate schedules and weight averaging are well known. However this is not emphasized in the paper.** I say this both in the sense that I knew them, and they are written up formally in [Sandler et al 2023](https://arxiv.org/abs/2301.02312) where the authors write:
> One practical result that we demonstrate that various popular averaging techniques have equivalent
learning rate schedules.

I recommend looking at Figure 1 of [Sandler et al 2023](https://arxiv.org/abs/2301.02312), where the authors show that a rapid learning rate decay at any point in training can give a big boost in performance to match the weight-averaged model. This can be done "any-time", and in my experience, many practitioners know this. From this perspective, it seems to me that Schedule-Free is likely finding a way to get the main iterates of the method to match the performance of the weight-averaged model directly, which is indeed really cool. However, the authors do not contextualize their result as such, since I think they are unaware of [Sandler et al 2023](https://arxiv.org/abs/2301.02312). This connection is also present in [Hägele et al 2024](https://arxiv.org/abs/2405.18392) who report that constant + cooldown schedule (basically the technique in Figure 1 of [Sandler et al 2023](https://arxiv.org/abs/2301.02312)) outperforms schedule free.

**The extent to which the paper is actually schedule-free is unclear** For the method to really be schedule-free, the optimal hyper parameters need to be invariant to the stopping time. Otherwise, tuning hyperparameters may implicitly be tuning a schedule. However, the authors do not really explore this. And furthermore, the experiments in the rebuttal suggest that this invariance is likely not the case. Looking at the bottom left plot, you see that the curves for different momentum values cross during training.

**There is no actual evidence provided for a connection between the theorems and the practical performance of the method** Remember that in science, for a theory to be falsifiable, it should make some kind of testable prediction. I'd like to see the authors produce a falsifiable prediction from their theory and then actually test it. Otherwise this aspect of the paper feels more like mathematical storytelling to me. I am doubtful about the usefulness of online convex optimization for analyzing deep learning algorithms.

Again, I don't mean to be gloomy, just I believe there should be a high bar for an award quality paper, and I invite the authors and other reviewers to respond.

---

> ### Author Response · Authors · 2024-08-10
> **Response**
>
> We are very glad to see the level of discussion here, it's rare to have so responsive reviewers! The points that Reviewer Q984 make are relevant, and so we will address them in detail:
>
> ## Novelty of weight averaging
> *The connections between learning rate schedules and weight averaging are well known*
>
> We discuss this at the beginning of our introduction, quoting from our work: ".. Zamani and Glineur (2023) and Defazio et al. (2023) showed that the exact worst-case optimal rates can be achieved via carefully chosen learning rate sequences (also known as schedules) alone, without the use of averaging. This result suggests that schedules have, in some sense, the same role to play as PR averaging in optimization". We tried to be careful to not over-claim here, we are not saying we discovered this link between averaging and schedules. Our claim is that our work is the first to give a method that uses averaging that *does not require a schedule*, while still matching or outperforming schedule-based approaches.
>
> The work of Sandler et al 2023 doesn't use averaging to yield a schedule free method. They show that for stochastic quadratic models, you can achieve the convergence rate by averaging $k$ weights as you can by reducing the learning rate by a factor $k$. A quote:
>
> “Thus averaging two solutions with a higher learning rate $\lambda$ gives us practically identical solution as if following the trajectory from $\theta_1$ to $\theta_2$, but with the learning rate $\lambda/2$.”
>
> There work is relevant and we will add a citation to our related work section, but their result is orthogonal to ours. They use averaging to replace the LR decreases in a regular schedule, but that still requires you to choose when to introduce and increase the amount of averaging, it's not "schedule-free", and the idea of the existence of schedule-free methods is not discussed in their work.
>
>
> ## Schedule-Free-ness
> *The extent to which the paper is actually schedule-free is unclear*
>
> We are running more experiments to verify this, as this is an important concern. Our theoretical results are very clear here, they explain why it's possible to learn without a time-dependence on the hyper-parameters. The experiments we include in the global PDF show that it may be possible to achieve a slightly higher test accuracy at the early stages of training by using less momentum, but the difference is tiny. We also show in Figure 1 in our work that a single run of Schedule-Free is competitive with runs with cosine schedules of varying lengths, which strongly supports our claim.
>
> ## Falsifiable Predictions
>
> *There is no actual evidence provided for a connection between the theorems and the practical performance of the method*
>
> This question is a difficult one to tackle, as we present a lot of empirical results to support our theory, but it's unclear to what degree these can be considered *falsifiable predictions*.
>
> Our work was motivated by the earlier theoretical results which suggested that averaging should be able to match schedules in performance; this could be considered the core theoretical prediction behind our work. In the introduction we state the question:
>
> "Do there exist iterate averaging approaches that match the empirical performance of learning rate schedules, without sacrificing theoretical guarantees?"
>
> This is not quite a falsifiable prediction in the classical sense of the *scientific method*, but we do demonstrate a method that matches the theoretical behavior predicted of averaging methods; this is strong empirical and scientific evidence.
>
> Our paper also introduces a new form of momentum, which has appealing theoretical properties not satisfied by regular momentum, such as being worst-case optimal for any choice of momentum parameter. We can generally expect that a method with better worst-case bound should perform the best, and that is what we show empirically: our method out-performs classical momentum. This is in a sense a prediction of the theory that is born out in practice.
>
> We hope reviewers will judge that our theoretical and empirical results are strong and convincing, even though we don't strictly follow the framework described by Reviewer Q984.
>
> --------------
>
> We are very glad to be able to present our work to responsive and thoughtful reviewers. We hope that our comments above will be considered on their merits.

---

> > ### Comment · Reviewer_Q984 · 2024-08-11
> >
> > Hi---I don't have a lot of time left for discussion as I'm neglecting other papers in my batch unfortunately.
> >
> > **Connection to [Sandler et al](https://arxiv.org/abs/2301.02312)**
> >
> > My point here is that if you look at Figure 1 in [Sandler et al](https://arxiv.org/abs/2301.02312)---**which I think the AC and all reviewers should do**---then it clearly suggests that there is performance being left on the table in conventional training, which I think your method is getting, and bravo for that. This provides a way to motivate your technique without any reference to online convex optimization. Just to be completely blunt, I do not trust the arguments you make about "exact worst-case optimal rates" and again I do not currently trust that your method is truly schedule-free.
> >
> > **Schedule-Free-ness**
> >
> > *"The experiments we include in the global PDF show that it may be possible to achieve a slightly higher test accuracy at the early stages of training by using less momentum, but the difference is tiny"*---thank you for acknowledging this. I encourage you to explore this aspect further, and am grateful that you are engaging with my critique
> >
> > **Falsifiable Predictions**
> >
> > I don't really have anything extra to add to what I already said, but thanks for engaging.
> >
> > **My score**
> >
> > I'm unwilling to raise my score while other reviewers are giving this paper a 10/10. I'm not sure what the appropriate reviewer etiquette is in this situation, so please feel free to flag this to the AC. **Also AC please read this**. Good luck!

---

### Decision · Program_Chairs · 2024-09-25

**Decision:**

Accept (oral)

**Comment:**

The reviewers are unanimously positive on this paper. They believe that the proposed approach has good potential to be quite significant for the field, both practically (as evidenced by good performance on AlgoPerf), and theoretically (it is a novel combination of momentum and weight averaging with theoretical analysis). They raised concerns about the hyperparameter sensitivity of the method and its relation to other work.

I agree with the general assessment on the potential impact of the paper and feel that it would be of very wide interest to the community. Therefore, I recommend to accept the paper as an oral.